# First Insights about Antiparasitic and Action Mechanisms of the Antimicrobial Peptide Hepcidin from Salmonids against *Caligus rogercresseyi*

**DOI:** 10.3390/pharmaceutics16030378

**Published:** 2024-03-08

**Authors:** Paula A. Santana, Camila Arancibia, Laura Tamayo, Juan Pablo Cumillaf, Tanya Roman, Constanza Cárdenas, Cinthya Paillan Suarez, Claudio A. Álvarez, Fanny Guzman

**Affiliations:** 1Instituto de Ciencias Aplicadas, Facultad de Ingeniería, Universidad Autónoma de Chile, Santiago 8910060, Chile; paula.santana@uautonoma.cl (P.A.S.); cinthya.paillan@cloud.uautonoma.cl (C.P.S.); 2Departamento de Química, Facultad de Ciencias, Universidad de Chile, Santiago 7800003, Chile; camila.arancibia.1@ug.uchile.cl (C.A.); laura.tamayo@uchile.cl (L.T.); 3CRC Innovación, Puerto Montt 5507642, Chile; jpnenen@gmail.com; 4Núcleo Biotecnología Curauma, Pontificia Universidad Católica de Valparaíso, Valparaiso 2373223, Chile; tanya.roman.b@mail.pucv.cl (T.R.); constanza.cardenas@pucv.cl (C.C.); 5Laboratorio de Cultivo de Peces, Departamento de Acuicultura, Facultad de Ciencias del Mar, Universidad Católica del Norte, Coquimbo 1781421, Chile; 6Laboratorio FIGEMA, Centro de Estudios Avanzados en Zonas Áridas, Coquimbo 1781421, Chile

**Keywords:** AMPs cysteine-rich, hepcidin, *Caligus* spp., antiparasitic, salmon peptide, chitin, fish skin

## Abstract

Currently, one of the primary challenges in salmon farming is caligidosis, caused by the copepod ectoparasites *Caligus* spp. The infection process is determined by the copepod’s ability to adhere to the fish skin through the insertion of its chitin-composed filament. In this study, we examined several antimicrobial peptides previously identified in salmonid mucosal secretions, with a primary focus on their potential to bind to chitin as an initial step. The binding capacity to chitin was tested, with hepcidin and piscidin showing positive results. Further assessments involving cytotoxicity in salmonid cells RTgill-W1, SHK-1, RTS-11, and RT-gut indicated that the peptides did not adversely affect cell viability. However, hemolysis assays unveiled the hemolytic capacity of piscidin at lower concentrations, leading to the selection of hepcidin for antiparasitic assays. The results demonstrated that the nauplius II stage of *C. rogercresseyi* exhibited higher susceptibility to hepcidin treatments, achieving a 50% reduction in parasitic involvement at 50 µM. Utilizing fluorescence and scanning electron microscopy, we observed the localization of hepcidin on the surface of the parasite, inducing significant spherical protuberances along the exoskeleton of *C. rogercresseyi*. These findings suggest that cysteine-rich AMPs derived from fish mucosa possess the capability to alter the development of the chitin exoskeleton in copepod ectoparasites, making them therapeutic targets to combat recurrent parasitic diseases in salmon farming.

## 1. Introduction

One of the first and most crucial immune barriers faced by fish pathogens is the epithelial mucosa, which contains components of innate immunity, such as glycoproteins, complement proteins, proteolytic enzymes, immunoglobulins, and antimicrobial peptides (AMPs), among others [1]. The latter are small peptide molecules, typically cationic and amphiphilic, playing a fundamental role in cellular and molecular defense against pathogens [2,3]. AMPs are associated with a broad range of functions, including broad-spectrum antimicrobial activity, meaning that they possess antibacterial, antiviral, antifungal, and antiparasitic properties. Additionally, they may exhibit immunomodulatory properties and cytotoxic activity against cancer cells [4].

The role of AMPs in host defense primarily involves interfering with the cellular mechanisms of pathogens due to their physicochemical properties [2]. Among these properties, some of the most relevant are length, charge, and hydrophobicity. Concerning length, AMPs are generally peptides containing between 12 and 50 amino acid residues. However, in the particular case of salmonid AMPs, their length varies from 11 to 79, with an average length of 41 amino acid residues [5]. This parameter can impact the peptide’s function differently. Although it has been demonstrated that a higher peptide length can contribute to increased microbial activity, in some cases, the fragmentation and proteolysis of long peptides can lead to higher antimicrobial activity compared to the activity presented prior to proteolysis [5]. An example of this is illustrated by cathelicidins, which are AMP produced as inactive precursors, which, when processed, release the active peptide [6,7]. Conversely, NK lysine, an active AMP of 74 amino acid residues, does not need to be fragmented to enhance its antimicrobial activity [8]. Regarding the net charge of salmonid AMPs, these molecules carry a positive charge ranging from 0 to +30. Histone H2A has the lowest net charge (0), while oncorrycin II has the highest net charge (+30); both peptides derive from rainbow trout [5,9,10]. Hydrophobicity, defined as the proportion of hydrophobic residues within the sequence, is typically around 50% for most AMPs, and it is suggested that this may have significant effects on the peptide’s function [11]. Although this is a crucial property for effective membrane permeabilization, it has been reported that both the proportion and distribution of polar and non-polar residues can lead to increased hemolytic activity [12,13], cell toxicity, and a loss of antimicrobial specificity due to changes in selectivity between eukaryotic and prokaryotic cells [14]. Therefore, this requires more in-depth study, as there are no conclusive results on the direct effects of hydrophobicity on hemolysis for salmonid AMPs [5,11]. This is evident with cathelicidins, wherein Cath1, which has 8% hydrophobicity, is extremely hemolytic for Atlantic salmon erythrocytes, while Cath2 showed no hemolytic activity, despite having 22% hydrophobicity [5].

Within AMPs are antiparasitic peptides (APPs), which are humoral response peptides produced during a parasite-mediated infection [15]. Concerning the mechanisms of action described for APPs, both extracellular and intracellular targets have been identified. In terms of extracellular targets, the destabilization of the parasite membrane has been documented. This effect is mainly due to the amphipathicity and cationicity of the peptides, interacting with the negatively charged external membranes of the pathogens [16,17]. Beyond damage to the membrane and cell surfaces, APPs exhibit intracellular action by entering the cytoplasm without causing noticeable and persistent damage to the parasite membrane. This process leads to cell death through interaction with intracellular targets [18]. It is noteworthy that these peptides must contain basic amino acid residues to be able to penetrate into the cell [19]. Examples of such peptides include histatin 5, an APP present in human saliva, which induces the depolarization of the plasma membrane of *Leishmania amazonensis*. This process allows the peptide to enter the cytoplasm and destabilize mitochondrial membranes [20]. Another example is magainin, a peptide that induces mitochondrial toxicity and apoptosis by activating Caspase-3/7 through the intracellular delocalization of calcium ions [21].

Up until now, research on AMPs present in fish epithelial mucosa has primarily focused on their antiviral and antibacterial activities, rather than on their role as APPs. A review has allowed the compilation of some fish AMPs that may exhibit antiparasitic activity, but their efficacy has only been validated against freshwater parasites such as *Ichthyophthirius multifiliis* or *Tetrahymena pyriformis* [22]. Therefore, further research is needed to advance in this direction, targeting the most significant pathogens for fish aquaculture.

Caligidosis is a significant health challenge in Chilean salmon farming, caused by the pathogen *Caligus rogercresseyi*, a copepod ectoparasite prevalent in highly productive geographical areas for salmon farming [23]. The infection process hinges on the pathogen’s ability, in its infective stage, to breach the fish skin mucus and attach to the dermis through the production of a chitin filament [24]. This attachment occurs during the transition from nauplius II to the copepodite phase, where the parasite settles on the host by pressing its cephalothorax against the fish’s skin, acting like a suction cup for initial adhesion. Subsequently, it firmly attaches using antennae and oral structures (maxilipeds) as hooks, allowing the insertion of a chitin filament that definitively anchors it to the fish. Following copepodite fixation, metamorphosis occurs into the chalimus stages, characterized by differences in body size, appendages, and an increasing number of chitin rings in their filament, reflecting progressive exoskeleton growth [25]. During this metamorphosis process, parasite development and growth rely on molting, a process necessitating the synthesis and recycling of chitin for the formation of the new exoskeleton [25]. In this context, one therapeutic target for antiparasitic compounds is the inhibition of chitin synthesis [26].

Until now, there have been no studies analyzing the antiparasitic capacity of AMPs present in the epithelial mucosa of salmonids against *C. rogercresseyi* ectoparasites. In this study, we assessed several AMPs previously identified in salmonid mucosal secretions for their ability to bind to chitin as an initial step. Subsequently, we evaluated the cytotoxicity of these compounds against salmonid cells. Finally, the least toxic molecule for the host cells was further investigated, determining its antiparasitic capacity against the nauplius II and copepodite stages of *C. rogercresseyi*, along with exploring the potential mechanism of action on the chitin exoskeleton.

This research represents the first study demonstrating the antiparasitic capacity of hepcidin against copepod ectoparasites. Consequently, their presence in the epidermal mucus of salmonids should be the focal point of future studies.

## 2. Materials and Methods

### 2.1. Acquiring Sequences of Peptides Derived from Salmonids with Antiparasitic Potential

A compilation was conducted in databases and literature to identify AMPs with evidence of broad-spectrum antimicrobial action from teleost fish, specifically focusing on the Salmonidae family. Peptides were selected from the main families of AMPs, including gramicidin, peptides derived from histones, piscidins, and hepcidins. Subsequently, peptides with a size lower than 30 amino acid residues and containing histidine, cysteine, serine, and aromatic residues were chosen. The selected peptides are presented in Table I, with their corresponding accession numbers in NCBI. These include gramicidin GsD (P69840.1), type histone H2B (XP_036804284.1), epinecidin 1 (APM86638.1), piscidin 2 (XP_035533992.1), and hepcidin (XP_021450828.1).

### 2.2. Peptide Modeling Using Bioinformatics Tools

An in silico analysis was conducted to derive the physicochemical properties of each selected peptide utilizing the Pepdraw platform (http://pepdraw.com/) accessed on 20 October 2021. Additionally, a prediction of the secondary structure for each peptide was carried out through homology modeling on the I-TASSER server (https://zhanggroup.org/I-TASSER/) accessed on 30 October 2021. In the case of hepcidin, the structure was obtained from Alvarez et al. [27].

### 2.3. Synthesis and Determination of Secondary Structure of Potentially Antiparasitic Peptides

The synthesis of the peptides was conducted via the Fmoc/t-butyl strategy technique in solid phase using the Tea-bag strategy [28,29]. In brief, Rink amide resin (Iris Biotech, Marktredwitz, Germany) with a substitution of 0.65 mmol/g was employed for peptide synthesis. Additionally, a portion of peptide-resin was employed for the coupling of rhodamine B.

Cleavage and final deprotection were carried out using a trifluoroacetic acid solution (TFA/H_2_O/triisopropylsilane/diethanethiol DOTA) (92.5:2.5:2.5:2.5, *v*/*v*/*v*/*v*) for 180 min at room temperature. Subsequently, the peptides were precipitated with diethyl ether, extracted with water, and lyophilized. The mass of the peptides was analyzed through electrospray ionization mass spectrometry (ESI-MS), and their purity was assessed using reversed-phase HPLC with a 0–70% acetonitrile-water gradient for 8 min at a flow rate of 1 mL/min.

The peptides were purified with Sep-Pak C-18 columns (Clean-Up^®^, United Chemical Technologies, Bristol, VA, USA) to be used in functional analysis.

The secondary structure of the synthesized peptides was determined using circular dichroism spectroscopy (CD). The Jasco J-815 Spectrometer equipment coupled with a Jasco CDF4265/15 Peltier temperature controller (Jasco Corp., Tokyo, Japan) was employed in the far ultraviolet (UV) range (190–250 nm), using quartz cuvettes with a path length of 0.1 cm. Peptide solutions were prepared at a concentration of 1.0 ± 0.5 mM in 30% trifluoroethanol (TFE 30%) in water. The molar ellipticity was calculated for each peptide, after subtracting the solvent blank. The resulting data were analyzed using Spectra Manager software (Version 2.0, JASCO Corp., Tokyo, Japan).

### 2.4. Chitin Binding Assay

For the chitin binding assay, magnetic chitin beads (New England BioLabs, Ipswich, MA, USA) were utilized, enabling the determination of their binding capacity of proteins and peptides [30,31,32]. The beads were pre-washed three times with binding buffer (20 mM Tris HCl (pH 8)/500 mM NaCl/1 mM EDTA/0.05% Tween-20). Subsequently, a solution of the peptide dissolved in ultrapure water (concentration 1 mg/mL) and binding buffer was added, and this mixture was incubated for 3 h with shaking at 100 rpm. Afterward, the supernatant was recovered, and the pellet containing the beads underwent three washes with binding buffer, with the recovery of the supernatant after each washing. To elute the bound peptide and remove the beads, elution buffer (20 mM Tris HCl [pH 8]/500 mM NaCl/1 mM EDTA/0.05% Tween-20/3 mM DTT) was used, with incubation and shaking at 100 rpm on ice overnight, followed by the recovery of the supernatant.

To validate the proper elution, the same experimental procedure for chitin binding was repeated, but this time using an elution buffer with a higher concentration of NaCl (20 mM Tris HCl (pH 8)/1M NaCl/1 mM EDTA/0.05% Tween-20/DTT 3 mM), yielding consistent results.

The eluates were analyzed on a 15% SDS-PAGE gel using the chitin-binding peptide mytichitin from *Mytilus coruscus* as the positive control [33], and a peptide derived from the carboxyl terminus of IL-8 (ssIL-8α) as the negative control [34].

### 2.5. Determination of Hemolytic and Cytotoxic Activity of Peptides

Hemolytic activity was assessed using red blood cells obtained from fresh salmon blood, according to Santana et al. [35]. For the assay, the erythrocyte solution was mixed in a 1:1 ratio with a peptide solution between 5 and 150 µM. PBS without the peptide served as a negative control (0% hemolysis), and 0.5% Triton X-100 in PBS was used as a positive control (100% hemolysis). These aliquots were incubated for 1 h with gentle shaking (100 rpm) at 20 °C.

The assays were conducted in triplicate, and the determination of hemolytic activity at different peptide concentrations was calculated using the following equation:$$\%Hemolysis=\frac{Abs\;peptide-Abs\;0\%\;lysis}{Abs\;100\%\;lysis-Abs\;0\%\;lysis}\times 100$$

For the determination of cytotoxicity, the cell lines used were: SHK-1, a macrophage-like cell line derived from the head kidney of Atlantic salmon (ECACC N° 97111106), RTgill-W, an epithelial-like cell line isolated from the gill of rainbow trout (ATCC CRL-2523), RT-gut, an epithelial-like shape from the intestine of rainbow trout [36], and RTS-11, a macrophage-like cell line of rainbow trout, initially derived from a prolonged hematopoietic culture of the spleen [37]. These cell lines were maintained at 18 °C in Leibovitz L-15 medium supplemented with 10% serum and antibiotics (penicillin-streptomycin). Cells were seeded in 96-well plates (50,000 cells/well) and incubated overnight for adhesion.

Cytotoxicity measurements were performed using a WST-1 (Merck/Sigma-Aldrich, Darmstadt, Germany) viability assay. Approximately 40,000 cells per well, contained in a volume of 200 μL of supplemented L-15 medium, were seeded in a 96-well plate at 18 °C for 24 h. Subsequently, the culture medium was removed, and two washings were performed with PBS. Then, 100 μL of a peptide solution in culture medium without serum or antibiotics was added at concentrations of 150, 75, 50, 25, and 5 µM, followed by incubation for 24 h at 18 °C.

After 24 h, the culture medium was removed, and 100 μL of a WST-1 dilution in culture medium without serum or antibiotics (10 μL of WST-1 in 100 μL of medium) was added. The plate was incubated without exposure to light for 4 h at 18 °C, and then the absorbance was read at 440 nm and 600 nm with a VERSA max microplate reader (Molecular Devices, LLC, San Jose, CA, USA). Cells in culture medium without serum or antibiotics were used as a negative control, while 0.1% Triton-X100 was used as a positive control. The assay was performed in triplicate.

The percentage of cell viability was calculated using the following equation:$$\%Viability:=\frac{Abs\;peptide-Abs\;positive\;control}{Abs\;negative\;control}$$

### 2.6. Antiparasitic Activity against C. rogercresseyi

#### 2.6.1. Larval Culture of *C. rogercresseyi*

Ovigerous females of *C. rogercresseyi* were collected from smoltified salmonids infected with the pathogen. Females were selected based on embryo development, and only those with dark pigmentation (ranging from brown to black) indicating an advanced stage of development, and thus close to eclosion, were used. Egg chains were cultivated in a conical tank with continuous water movement facilitated via bottom aeration. Upon eclosion, larvae were separated from non-eclosed chains and cultivated until utilized in bioassays. The nauplius used were obtained on the same day of eclosion, and their nauplius stage were confirmed through optical microscopy [38].

To obtain copepodites, larvae were maintained for four days in 5 μm filtered seawater at 11 ± 1 °C, 32 PSU, 12:12 h photoperiod, and a culture density of 1–2 larvae/mL. Dead parasites were removed from the tank’s bottom, and their stage was confirmed through optical microscopy.

#### 2.6.2. In Vitro Antiparasitic Activity against *C. rogercresseyi*

In vitro determination of antiparasitic activity against *C. rogercresseyi* through sensitivity bioassays involved grouping nauplius and copepodites with good shape through separation using a 120 μm mesh screen with a reduced volume of seawater. This parasite stock was distributed in 96 flat-bottom wells using a mini-screen with the assistance of a micro-spatula.

The sensitivity bioassay methodology with parasite larvae aims to ascertain the percentage of individuals affected and/or deceased at each concentration of the chemotherapeutic compound.

The antiparasitic effect of peptides was evaluated at concentrations of 50, 75 and 150 µM in seawater, inoculating 200 µL in each of the 96 wells, while seawater was used as a negative control. Bioassays and plate maintenance were conducted at 12 °C, and the effects evaluated in eight replicates, according to a previous procedure.

Readings were taken every 24 h. The inhibition of parasite movement was evaluated through classifying them as alive, dying, or dead. Alive: nauplius and copepodites larvae swim actively with constant impulses, mostly in the water column. Dying: slow movement, weak and erratic impulses, unable to remain in the water column for an extended period; in a critical condition, only appendages, organs and digestive system movement can be appreciated. Dead: No movement of appendages, organs or the digestive system. “Affected” refers to the sum of those dying and dead. Larvae were exposed for up to 144 h at different peptide concentrations.

Larvae classified as dying or dead were examined using a stereoscopic microscope Euromex NexiusZoom model NZ.1903P (Carl Zeiss, Oberkochen, Germany).

### 2.7. Fluorescence and Scanning Electron Microscopy

For fluorescence microscopy, the nauplius parasites were fixed with 2.5% glutaraldehyde in filtered seawater for 2 h at room temperature. Subsequently, the samples were dewaxed using neoclear and then hydrated using a descending gradient of ethanol (100, 96, and 70%), culminating with distilled water. Following hydration, the samples were mounted using DAPI Fluoromount-G. The slides were then placed in a horizontal position for drying in the dark for 24 h at room temperature before imaging. In this study, fluorescent images were captured with an Automated Microscope and Live Cell Imager, BioTek Lionheart FX. Confocal images were processed and generated using Imaris software (V 7.4.2, ImarisX64; Bitplane AG, Schlieren, Switzerland)

For scanning electron microscopy (SEM), the fixed nauplius parasites were dehydrated using increasing concentrations of ethanol (30, 50, 70, 80, 90, and 100%) and coated with a thin gold film. Finally, the samples were observed using a Hitachi TM3000 microscope operated at 15 kV.

### 2.8. Statistical Analysis

The data were subjected to analysis for normality and homoscedasticity in each study using the Shapiro–Wilk and Fligner–Killeen tests, respectively. To compare the antiparasitic effects of the peptides and doses evaluated, a two-way analysis of variance (ANOVA) was conducted, followed by a comparison of means using the Tukey test. In the case of cytotoxicity and hemolysis analyses, a one-way ANOVA was employed, followed by a comparison of means using the Tukey test. Statistically significant differences were considered at *p* < 0.05.

All data were analyzed and graphed using GraphPad Prism 8.1 statistical software.

## 3. Results

### 3.1. Synthesis and Characterization of Salmonid AMPs with Potential Antiparasitic Activity

A database search for broad-spectrum AMPs described for the Salmonidae family yielded 63 potential peptides. Subsequently, peptides present in mucosa were selected, narrowing the list down to 24 peptides. Some of them have proven antiparasitic activity, as seen in fish from the Moronidae family, while others have not been explicitly described with antiparasitic activity. However, that possibility has not been ruled out. Based on these results, peptides composed of up to 30 amino acid residues were selected according to the experimental strategy using chemical synthesis. Selected peptides contained amino acids (tyrosine, tryptophan, glycine, cysteine, histidine, and/or serine) known to be involved in binding to the polymer chitin. The final selection included AMPs belonging to the gramicidin, peptides derived from histones, piscidins and hepcidins families. The detailed sequence and physicochemical properties of the selected peptides are presented in Table 1.

All the selected peptides have a net positive charge at pH 7.0, with the H2B peptide having the lowest positive charge (+1) and the highest hydrophobicity. The hepcidin peptide has the greatest number of cysteines in its sequence (eight residues), followed by epinecidin (seven residues), while piscidin has the highest number of histidine residues [4].

Three dimensional modeling was carried out on the I-TASSER server to obtain the putative secondary structure of the peptides, with the results shown in Appendix A. The peptides GsD, epinecidin and piscidin showed an α-helix-type structural propensity, while the peptide H2B presented a mainly random coil structure.

With this background information, the selected AMPs were chemically synthesized. The peptides were further characterized by HPLC and mass spectrometry, as detailed in Table 2 and Appendix A. The secondary structure determined using CD was consistent with the structures modeled through I-TASSER. The CD spectra for GsD and piscidin showed characteristic features of an α-helix-type structural motif: two minimum molar ellipticity at 200 and 230 nm, and a maximum molar ellipticity between 180 and 200 nm (Table 2). Epinecidin displayed a less defined CD spectrum, indicating a mixed structure, while H2B showed a tendency toward a random coil structure.

These comprehensive characterizations provide a foundation for the subsequent evaluation of the peptides’ antiparasitic activity.

### 3.2. Evaluation of the Chitin-Binding Capacity of Salmonid AMPs

This study aims to determine the capacity of AMPs from salmonid mucosa to affect the viability of marine ectoparasites with a chitin exoskeleton, such as *C. rogercresseyi*. Initially, the ability of selected AMPs to bind to chitin was assessed.

As shown in the SDS-PAGE gels in Figure 1, after the washing and subsequent elution processes, it is evident that the ssIL-8 peptide lacks the ability to bind to chitin (Figure 1a). As expected, the mytichitin peptide is visible in the elution stage (Figure 1b). Only two AMPs selected for this study, piscidin and hepcidin, demonstrated binding to chitin, as evidenced by their presence in the first washing step and the last elution step (Figure 1c,d, respectively). Additionally, the intensity of the hepcidin eluate band appears greater than that of piscidin, and even exceeds that of the mytichitin control. In the case of peptides H2B, GsD, and epinecidin, their presence cannot be observed in both the washing and elution steps (Figure 1e–g).

Based on these results, the study proceeded with the peptides piscidin and hepcidin.

### 3.3. Determination of the Cytotoxic Activity of AMPs with Antiparasitic Potential

The potential use of AMPs in combating infectious diseases relies on ensuring that these compounds are nontoxic to host cells at defined concentrations for therapeutic purposes. Therefore, the cytotoxic activity of hepcidin and piscidin was assessed in erythrocytes of *Salmo salar*, as well as in cell lines derived from mucosal cells and leukocytes of *Oncorhynchus mykiss* and *S. salar*.

The results indicate a significant hemolytic activity of the piscidin peptide, leading to 50% hemolysis at concentrations exceeding 15 µM (Figure 2a). Conversely, hepcidin demonstrated less than 50% hemolysis at all concentrations analyzed (Figure 2b).

In cytotoxicity assays on cell lines both peptides between 5 and 150 µM, exhibited no detrimental effect on the viability of the trout cell lines RTS-11, RT-gut, RT-gill and SHK-1 (Figure 2c–j).

Due to the observed high hemolytic activity of piscidin, the decision was made to proceed with the evaluation of the antiparasitic activity of hepcidin against *C. rogercresseyi*.

### 3.4. Effect of Hepcidin on the Viability of C. rogercresseyi

The antiparasitic activity of hepcidin against copepod ectoparasites was assessed in the nauplius II and copopodite stages of *C. rogercresseyi*, both characterized by chitin polymer-forming activity. As depicted in Figure 3, the impact of hepcidin treatment on parasites was much more pronounced in the nauplius II stage than in the copepodite stage, and its activity in both stages exhibited dose-dependency. Additionally, a time-dependent effect was observed in both stages, indicating that prolonged exposure led to an increased antiparasitic activity of hepcidin. Concentrations below 50 µM did not significantly affect parasite viability in either stage.

In the nauplius II stage, hepcidin affected 50% of the parasites after 24 h of incubation at a concentration of 50 µM, significantly enhancing its antiparasitic activity with prolonged exposure. Moreover, at concentrations exceeding 75 µM, no viability was observed (Figure 3a).

Conversely, the effects on the copepodite stage only became apparent after 48 h of exposure to the compound, with values exceeding 50% observed only at the highest tested concentration (150 µM) (Figure 3b).

### 3.5. Location and Morphological Changes Induced by Hepcidin in C. rogercresseyi Nauplius II Stage

The antiparasitic study revealed that hepcidin exhibits higher activity against the nauplius II stage of *C. rogercresseyi*. Consequently, it was decided to further investigate the potential mechanism of action of this peptide, focusing initially on its demonstrated binding capacity to chitin polymers. To facilitate compound detection, the peptide was re-synthesized with the rhodamine B fluorophore coupled to the amino terminus. This labeled hepcidin was used to locate the peptide within parasite tissues. Histological sections obtained from parasites treated with or without the peptide were examined using fluorescence microscopy to detect the rhodamine B signal (Figure 4).

Given hepcidin’s capability to interact with chitin polymers on the surface of parasites, it was analyzed whether the peptide could alter the exoskeleton of *C. rogercresseyi* in the nauplius II stage using scanning electron microscopy (SEM). After 24 h of incubation with hepcidin or a saline solution (control), various micrographs were obtained (Figure 5).

Figure 5A illustrates the morphology of control *C. rogercresseyi* in the nauplius II stage, displaying two main functional regions: the anterior cephalothorax and the posterior genitoabdomen. Both regions appear dorsally flattened and possess a smooth body shape typical of the exoskeleton. Additionally, typical antennular antennae of the nauplius stage are observed. Figure 5B–D provide higher magnification (800×) views of the dorsal regions, revealing the preserved surface of the parasite. Elongated and pointed structures on the surface correspond to needle-shaped crystalline formations from the sea salt used in the sample fixation treatment.

Conversely, Figure 5E depicts a parasite in the nauplius II stage treated with 75 µM of hepcidin, showing the lateral area of the anterior cephalothorax and the posterior genito-abdomen with spherical structures (protuberances) not typical of the parasite along the entire length of the exoskeleton. Figure 5F–H showcase the surface of the parasite at a higher magnification (500×), clearly revealing the presence of protuberances with a spherical morphology of variable sizes, ranging from the smallest with a diameter close to 4.6 µm to the largest measuring 18.5 µm.

## 4. Discussion

Fish mucosal organs, such as the skin, gills, and intestine, serve as continuous interfaces interacting with aquatic microbiota and the external environment, making them crucial entry routes for pathogens. Currently, copepod ectoparasites are particularly significant in the Chilean salmon industry [39,40]. The limited efficacy of agrochemicals and synthetic antiparasitic agents against these pathogens necessitates the exploration of novel strategies to combat ectoparasite spread in farming centers.

In this context, despite the extensive documentation of AMPs in fish mucosal secretions and their well-established antiviral and antibacterial roles [41,42], investigations into their antiparasitic role, especially against copepod parasites, are scarce. This scarcity is exacerbated by copepods possessing a chitin exoskeleton that protects them during various stages of their life cycle in aquatic environments [25]. The present study proposes that the ability to bind to chitin could be a discriminating factor for AMPs produced by fish, indicating their potential to interact with copepod ectoparasite exoskeletons. Our findings affirm that peptides with a higher abundance of cysteines are more effective in interacting with chitin polymers.

Thus far, peptide models for chitin binding have been mainly derived from hevein family peptides found in plants. The hevein-type mO1 and mO2, distinguished by their 6 to 10 cysteine residues, actively participate in the defense against fungal phytopathogens possessing chitin cell walls [43,44]. Hevein mO1 and mO2, featuring up to five intramolecular disulfide bridges, provide structural stability for interacting with the chitin structures of pathogenic fungi [45]. This structural characteristic is also observed in the model peptide mytichitin-1 isolated from the mollusk *M. coruscus*, which possesses three intramolecular disulfide bridges and a putative chitin-binding domain [33], validating its ability to bind to chitin. In the case of salmonid hepcidin, with its sequence composed of eight cysteines forming four disulfide bridges, it demonstrated a higher binding capacity to chitin polymers. Additionally, the well-defined β-sheet secondary structure, along with the presence of the hydrophobic amino acids phenylalanine and tryptophan in the structure’s loops, further supports its interaction with chitin, as reported for other chitin-binding peptides [31].

Regarding piscidin, despite the absence of cysteines in its sequence, the high content of histidines suggests a potential mediation in chitin binding. Histidine’s imidazole ring in its side chain is a key residue in various binding domains in proteins and peptides [46], supporting its anticipated involvement in binding to polysaccharide structures like in chitin. Therefore, our findings indicate the absence of a direct correlation between the structure of AMPs and their binding capacity with the polymer. Instead, it is the presence of specific amino acids, such as cysteines and histidines, that plays a crucial role in enhancing the stability of the binding and interaction motif with the chitin polymer. Consequently, these amino acids should be taken into consideration when designing or improving peptides to achieve stronger interactions with chitin.

While inherently non-quantitative, chitin-binding assays have conventionally utilized chitin beads for protein and peptide purification, followed mainly by assessments of antifungal activity [47,48,49]. Our research goes beyond the typical applications of chitin beads, demonstrating that hepcidin showcases a heightened binding capacity to the chitin polymer of copepod parasites. Subsequent investigations on *C. rogercresseyi* have validated this efficacy, with results indicating that the nauplius II stage exhibits greater sensitivity to hepcidin’s action. This increased sensitivity might be attributed to the greater remodeling of the chitin polymer during the metamorphosis from the nauplius to the copepodite state. Remarkably, there are no works mentioning or describing peptides acting on the exoskeleton of fish parasites. The closest studies involve synthetic compounds known as benzoylurea-based chitin synthesis inhibitors (CSIs). These CSIs have demonstrated efficacy against the salmon louse (*Lepeophtheirus salmonis*), a closely related ectoparasite affecting the salmon industry in the northern hemisphere. CSIs can impact the molting of the chitin skeleton in the nauplius II stage, inhibiting metamorphosis to copepodite in a dose-dependent manner [50].

It is proposed that chitin synthesis inhibitors (CSIs) interfere with chitin formation in the procuticle and epicuticle deposition, resulting in abortive molt and death. The mechanism by which CSIs act on copepod parasites remains unclear, but it is hypothesized to involve key enzymes in the polymerization process rather than direct chitin binding, as observed in the present study with hepcidin.

It has been demonstrated that hepcidin is localized on the surface of the parasite, suggesting its antiparasitic action on the exoskeleton. The formation of spherical structures of varying sizes across the parasite’s surface, some nearing 20 µm, may result from an alteration in the molting process due to peptide binding, causing chitin polymer accumulation on the surface. In control parasites, the molting process presents rectangular and smaller spherical structures than those observed with the peptide treatment. These structures possibly separate from the parasite, potentially giving rise to new polymers shaping the subsequent copepodite stage. Chitin is present in the peritrophic matrix of these ectoparasites, covering the microvilli of their intestines [51], suggesting that effects on chitin could induce morphological changes in the intestine, influencing feeding behavior [52].

The antiparasitic effect of hepcidin on *C. rogercresseyi* appears related to its chitin-binding ability, causing alterations in key anatomical structures of the parasite. Although the present study provides an initial insight into the antiparasitic mechanism of cysteine-rich antimicrobial peptides, further research is required to elucidate the formation of observed spherical structures when facing peptides like hepcidin. The absence of similar functional studies on copepods opens a new avenue for exploring the development of antiparasitic strategies.

For peptides to be applicable, demonstrating antiparasitic activity with minimal cytotoxicity and hemolysis towards host cells is essential [53]. The present analysis shows that performing cytotoxicity assessments solely with fish cell lines is insufficient, as none of the peptides tested here exhibited negative effects on cell viability. However, piscidin displayed high toxicity towards salmon erythrocytes.

Hemolytic AMPs typically contain amino acids, such as tryptophan, lysine, and arginine, affecting hydrophobicity and/or peptide charge [54]. The lytic action of peptides with these amino acid residues is mediated by direct interactions with different lipid groups of cell membranes. Particularly, it is indicated that tryptophan residues play a fundamental role during the lysis of erythrocytes, as they are involved in the binding of peptides to the cholesterol of biological membranes [54]. Additionally, certain hemolytic AMPs, such as bombinins, magainins, and dermaseptins, exhibit considerable variation in terms of chain length, hydrophobicity, and charge. However, they share the characteristic of being linear lysine-rich cationic structures that adopt an α-helix conformation [55]. Observing these parameters in the analyzed peptides, the presence of tryptophan, lysine, and arginine in their amino acid sequence corresponds to 14% in piscidin (3 residues out of a total of 22) and 20% for hepcidin (4 residues out of a total of 20). Therefore, the possible explanation for piscidin being more hemolytic may be related to its secondary structure; an α-helix in the case of piscidin versus a coiled-coil or helical-loop-helix structure in the case of hepcidin. Experimental evidence has shown that helical peptides are significantly more lytic than those adopting a β-sheet structure, suggesting that structural elements also contribute to the lysis of fish erythrocyte membranes [11].

Fish AMPs hold promising potential for supplanting conventional antiparasitic agents, offering significant advantages to the aquatic environment. Present results will pave the way for continued exploration of the application of hepcidin in vivo antiparasitic studies. This will enable the demonstration of its antiparasitic potential in salmonids in the future.

## 5. Conclusions

The presence of cysteine and histidine in the AMPs composition from salmonid correlates positively with increased efficacy in interacting with chitin polymers, underscoring the importance of this structural feature. The differential sensitivity of developmental stages of *C. rogercresseyi*, particularly the nauplius II stage, suggests that hepcidin’s efficacy may be related to chitin polymer modifications during metamorphosis. The results support the importance of considering specific structural features of AMPs in designing antiparasitic strategies for salmon farming. However, further investigations are warranted to unravel the functional mechanisms of salmonids’ hepcidin, paving the way for future studies. Additionally, the enhancement of antiparasitic activity through peptidomimetics represents an avenue for refinement and innovation in the field.

## Figures and Tables

**Figure 1 pharmaceutics-16-00378-f001:**
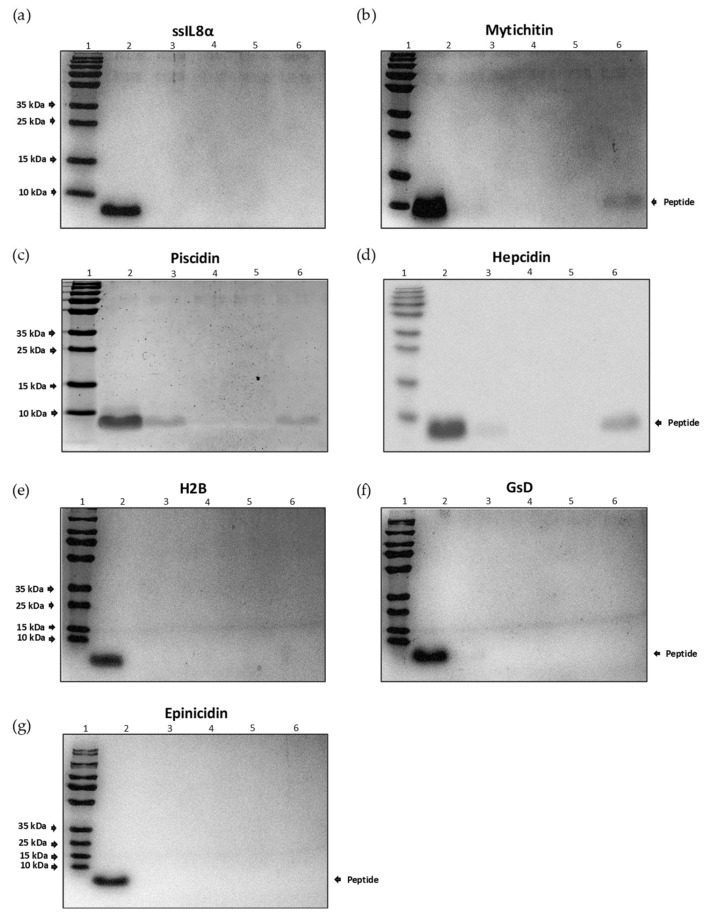
Chitin binding ability of AMP from salmonid mucus. Visualization of potential chitin-binding peptides. Images correspond to 15% SDS-PAGE–Coomassie blue stain: Lane 2: peptide not bound to chitin beads; Lane 3: washings for the recovery of unbound peptide; Lane 6: elution of peptide bound to chitin. The corresponding peptide is indicated at the top of each gel. The right arrow on each gel indicates the molecular mass band of each peptide (Lane 1). The letters (**a**–**g**) correspond to SDS-PAGE geles for peptides carboxyl terminus of IL-8 (ssIL8α), mytichitin, Piscidin, Hepcidin, type histone (H2B), gramicidin (GsD), and Epinicidin, respectively.

**Figure 2 pharmaceutics-16-00378-f002:**
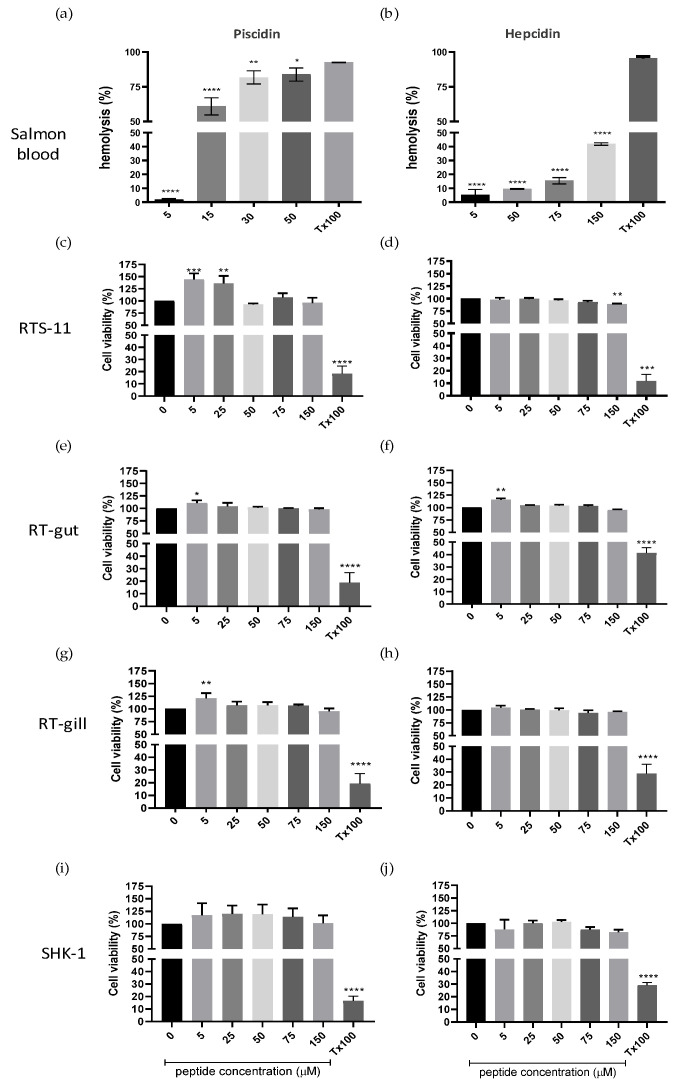
Cytotoxicity activity of piscidin and hepcidin in salmonid cells. Hemolytic activity was determined in salmon red blood cells at 5–50 µM concentration for piscidin (**a**) and 5–150 µM for hepcidin (**b**). PBS was used as a negative hemolytic control, while 0.5% Triton X-100 was used as a positive hemolytic activity. Cytotoxic effect of piscidin and hepcidin on RTS-11 cell line (**c**,**d**); RT-gut cell line (**e**,**f**); RTgill-W1 (**g**,**h**); and SHK-1 cell line (**i**,**j**). Cells were incubated for 24 h in the presence of different concentrations of piscidin and hepcidin (5–150 µM). Cell viability was measured using WST-1 assay. Data are expressed as mean ± SD of experiments performed in triplicate (* *p* = 0.0139; ** *p* = 0.0079; *** *p* < 0.005; **** *p* < 0.0001).

**Figure 3 pharmaceutics-16-00378-f003:**
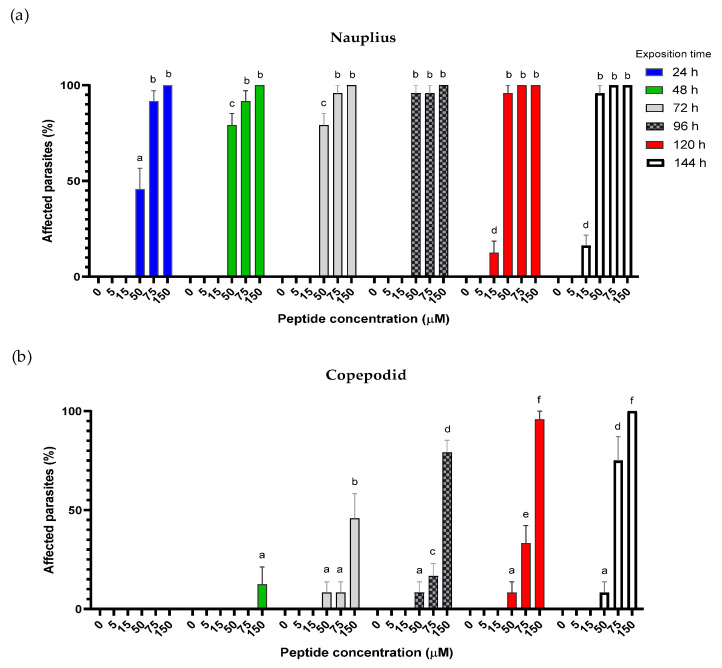
Antiparasitic activity of hepcidin against *C. rogercresseyi*. The graph shows the percentage of affected parasites against different concentrations of hepcidin (5–150 μM). Seawater was used as a negative control. The effect against nauplius (**a**) and copepodid (**b**) stages was evaluated between 24 and 144 h. Data are expressed as mean ± SD of experiments performed in triplicate. Significant differences are indicated with different letters with a *p* < 0.05 in each graph.

**Figure 4 pharmaceutics-16-00378-f004:**
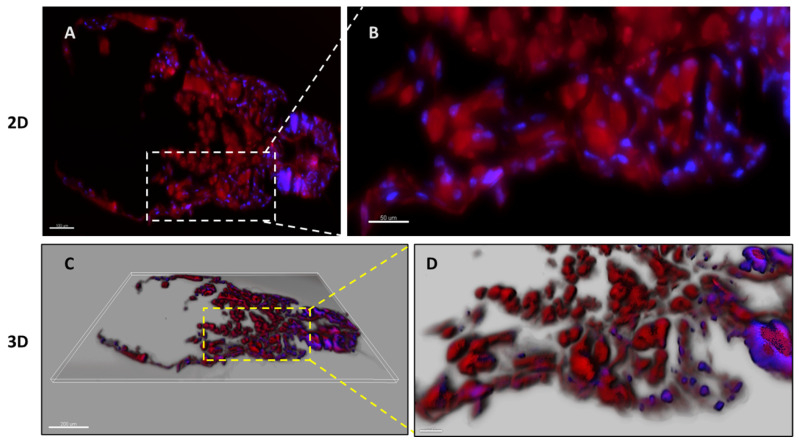
Location of hepcidin in *C. rogercresseyi* nauplius II. Fluorescence confocal microscopy analysis of the binding of hepcidin rhodamine labeled (red) at 150 μM to nauplius II stage. Parasite nuclei were stained with DAPI. The white and yellow segmented lines indicate the zoom zone images. (**A**) Single-plane fluorescence images captured from nauplius II stage at a magnification of 20×, with a measurement bar of 100 µm. (**B**) Magnified 40× single-plane fluorescence images depicting a closer view of the interaction between hepcidin and nauplius II, with a measurement bar of 50 µm. (**C**) Imaris 3D reconstruction of nauplius II at a magnification of 20×, with a measurement bar of 200 µm. (**D**) A 40× magnification of the 3D reconstruction illustrating the interaction between Hepcidin and nauplius II, with a measurement bar of 50 µm.

**Figure 5 pharmaceutics-16-00378-f005:**
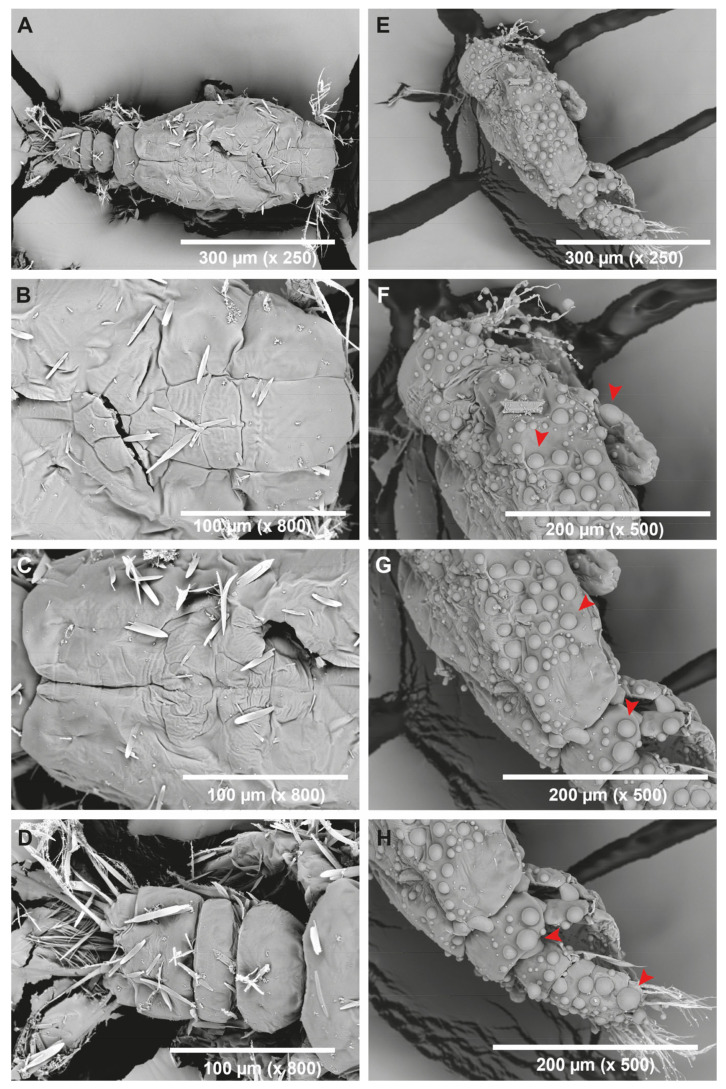
Surface damage of *C. rogercresseyi* nauplius by hepcidin. Scanning electron microscopy (SEM) micrographs of nauplius II. (**A**–**D**) correspond to micrographs of nauplius treated only with seawater. Micrographs (**E**–**H**) correspond to micrographs of nauplius treated with hepcidin at 75 µM. The red arrow indicates the presence of vesicle-like structures on nauplius II after treatment with hepcidin. The white bar at the bottom of each image indicates the size in µm.

**Table 1 pharmaceutics-16-00378-t001:** Identification and physicochemical characterization of AMPs with potential antiparasitic activity.

Peptide	MW		Retention Time (min)	CD Spectrum
	Theorical	Obtained (M + H)		
GsD	1543.9	1543.6	4.930	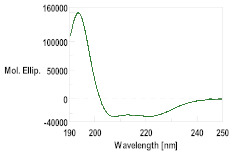
H2B	1593.8	1593.2	1.720	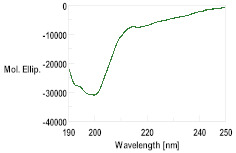
Epinecidin	2334.3	2334.3	8.563	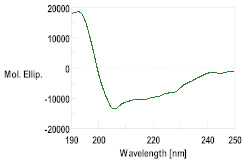
Piscidin	2542.4	2542.2	8.473	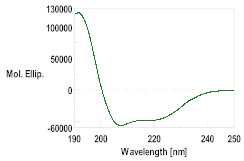
Hepcidin	2326.7	2327.4	13.205	β-sheet conformation (oxidized form) α-helix conformation (reduced form) [31]

**Table 2 pharmaceutics-16-00378-t002:** Characterization of synthetic AMPs with potential antiparasitic activity.

Peptide	Sequence	Fish Specie	Antiparasitic Activity	Mucosa	MW (g/mol)	pI	Net Charge (pH 7)	Hydrofobicity (Kcal/mol)	Genbank N°
GsD	FIGGIISFFKRLF	*Grammistes sexlineatus*	Not evaluated	Skin	1543.89	11.53	+2	+3.82	*P69840.1*
H2B	VSEGTKAVTKYTSSK	*Oncorhyn-chus mykiss*	*Saprolegnia* spp.	Gill/skin	1593.80	9.69	+1	+21.61	*XP_036804284.1*
Epineci-din	GFIFHIIKGLFHAGKMIHGLV	*Epinephelus coioides*	*Trichomonas vaginalis*	Gill/skin	2334.31	10.65	+2	+12.35	*APM86638.1*
Piscidin	FFHHIFRGIVHVGKTIHRLVTG	*Morone saxatilis*	*Cryptocaryon irritans.* *Trichodina* sp. *Amyloodinium ocellatum. Chthyophthirius multifiliis*	Gill/skin	2542.44	11.69	+3	+17.46	*XP_035533992.1*
Hepci-din	LCRWCCNCCHNKGCGFCCKF	*Oncorhyn-chus mykiss*	Not evaluated	Intestine/ skin	2326.86	8.08	+3	+14.72	*XP_021450828.1*
Peptide	Sequence	Fish specie	Antiparasitic activity	Mucosa	MW (g/mol)	pI	Net charge (pH 7)	Hydrofobicity (Kcal/mol)	*Genbank N°*
GsD	FIGGIISFFKRLF	*Grammistes sexlineatus*	Not evaluated	Skin	1543.89	11.53	+2	+3.82	*P69840.1*
H2B	VSEGTKAVTKYTSSK	*Oncorhyn-chus mykiss*	*Saprolegnia* spp.	Gill/skin	1593.80	9.69	+1	+21.61	*XP_036804284.1*
Epineci-din	GFIFHIIKGLFHAGKMIHGLV	*Epinephelus coioides*	*Trichomonas vaginalis*	Gill/skin	2334.31	10.65	+2	+12.35	*APM86638.1*
Piscidin	FFHHIFRGIVHVGKTIHRLVTG	*Morone saxatilis*	*Cryptocaryon irritans.* *Trichodina* sp. *Amyloodinium ocellatum. Chthyophthirius multifiliis*	Gill/skin	2542.44	11.69	+3	+17.46	*XP_035533992.1*
Hepci-din	LCRWCCNCCHNKGCGFCCKF	*Oncorhyn-chus mykiss*	Not evaluated	Intestine/ skin	2326.86	8.08	+3	+14.72	*XP_021450828.1*

## Data Availability

The data presented in this study are available in this article (and Appendix A).

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
