# Peer review of "First Insights about Antiparasitic and Action Mechanisms of the Antimicrobial Peptide Hepcidin from Salmonids against Caligus rogercresseyi"

_pharmaceutics, 2024, doi:10.3390/pharmaceutics16030378_

Round 1
Reviewer 1 Report
Comments and Suggestions for Authors
This study tried to address an infection problem in salmon farming caused by Caligus spp. Because of the chitin-composed filament playing a key role in the infection process , the authors focused on investigating the binding of some AMPs previously identified in salmonid mucosal secretions with chitin, and identified two AMPs- hepcidin and piscidin, but specifically selected hepcidin for antiparasitic assays in detail. The results showed that this AMP was active particularly in the nauplius II stage of C. rogercresseyi with supporting data of the active location of the AMP on the surface of the parasite. Because of hepcidin containing a higher number of cysteines, the authors concluded that cysteine-rich peptides and their structural features are important to combat the salmon farming infection caused by Caligus spp. The followings are some concerns and suggestions for the authors’ consideration:
- Chitin-binding assay was conducted for selected AMPs which showed some positive results, however, there are no quantitative results about the binding affinity.
- In the chitin binding assay, all peptide concentrations were 1mg/ml, however, peptides had different molecular weights due to their different numbers of amino acids in the sequences. That indicated they had different molar concentrations (uM or mM). However, the binding affinity depends on the molar concentration. Did the authors consider the concentration effect? Also, in Figure 1g, the protein band (Epinicidin) looks weaker than the protein bands in other gels.
- There is a gel labeling issue in Figure 1- all in Lane 1 are the protein marker, not ‘peptide not bound to chitin beads’. So, it should change ‘Lane 1’ to ‘Lane 2’, ‘Lanes 2-4’ to ‘Lanes 3-5’, ‘Lane 5’ to ‘Lane 6’.
- Line 184, ‘mitiquitin-CB’ should be ‘mytichitin-CB’
- Line 327, ‘DC spectrum’ should be ‘CD spectrum’
- In Table 2, CD spectra were provided for all peptides except for Hepcidin, which only showed citation. It is suggested to include some structural information from the reference, like helix or sheet, etc.
- The authors showed some structural information about the peptides, however, there is no further information about how these structures are related to their functions, at least for the chitin binding.
- Lines 371-372, it states ‘with the exception of hepcidin in the RTS-11 cell line, where viability was reduced by up to 90% at a 150 uM concentration (Figure 2D)’. However, the figure (Figure 2D) does not show the changes at the concentrations from 0-150 uM.
- In Figure 5, suggest changing the color of A-H on each image from black to white, especially for Image F, G, H, they are too dark for view.
- Only one cysteine-rich peptide was tested in this study, however, the authors concluded a higher number of cysteines would be important to be developed as antiparasitic agents. This seems not to be very convincing.
Reviewer 2 Report
Comments and Suggestions for Authors
The authors wonder if previously obtained antimicrobial peptides have anti-parasitic activity against the ectoparasitosis of Caligus sp, which is the cause of affectation of salmon production in regions of Chile. In this field the authors analyze the properties of antimicrobial peptides. (hepcidin, piscidin) as its mechanism of action on the alteration of chitin as part important development in fish and different stages of development (larva phase) and free swimming) by already established methods such as electron microscopy and fluorescence, in addition to others particularities of these peptides such as degree of cytotoxicity on fish cell line models, binding to chitin, concentrations among others.
In the current presentation there are certain important points that the authors should consider to improve the presentation of the manuscript.
In the material and methods section
Section 2.5 page 4, line # 198, related to cytotoxicity assays on fish cell lines, it is significant that the authors describe the origin of these lines, if applicable, cite a bibliographic reference, that is, are they of primary origin, epithelial cells? ??.
In this same section line #211-214 it is important to document the specifications of the WST reagent was commercial kit, the salt or cite a reference of the method. Also, include the specifications of the reader equipment used.
In the results section line #311, complete the abbreviation of CD referred to circular dichotrism spectroscopy (CD). It does not appear in previous material and methods text, for this reason it is suggested to introduce it in the text the first time it is referred to.
In the same section of results of the chitin analysis line #332 to 336 corresponds to the description of the chitin binding test and the basis of this test, I consider that the authors should review because it is not in accordance with the results, this description is directed regarding material and methods, I suggest eliminating lines # 332 to 336. Also, rephrase these paragraphs to give clarity to the description of the result in Figure 1.
section Figure 4 line # 444 to 448 related to the localization of hepcidin in the ectoparasite of C. rogercresseyi phase II increase the sharpness of the fluorescence image and the size of the bar indicating the magnification of each image does not is adequately appreciated.
In the same result section Figure 5, line #450 to #457, the bar that indicates the magnification size of the image is also not visible. I suggest changing the color of the bar and increasing the size of the numbers that indicate the magnification. Also include arrowheads or markers that indicate in the image the critical changes of the treatment with hepcidin, this change should be noted in the legend of the figure 5.
In the Discussion section line #477 to 481 clearly break down the pathway theory of chitin binding components mO1 and MO2, rephrasing is not clear in the current presentation.
Round 2
Reviewer 1 Report
Comments and Suggestions for Authors
Concerns are addressed either in the revision or with a plan for future study.
Reviewer 2 Report
Comments and Suggestions for Authors
Comments to the authors
The new version sent improved the impact that the authors demonstrate in their data obtained, in addition to the attention to the suggested observations.